# Early Embryonic Development in Agriculturally Important Species

**DOI:** 10.3390/ani14131882

**Published:** 2024-06-26

**Authors:** Fuller W. Bazer, Gregory A. Johnson

**Affiliations:** 1Department of Animal Science, Texas A&M University, College Station, TX 77843-2471, USA; 2Department of Veterinary Integrative Biosciences, Texas A&M University, College Station, TX 77843-2471, USA; gjohnson@cvm.tamu.edu

**Keywords:** embryonic development, pregnancy recognition signaling, implantation, pig, sheep, cow, horse

## Abstract

**Simple Summary:**

The period of early embryonic development in pigs, sheep, cows, and horses is critical, as two-thirds of embryonic losses occur during this peri-implantation period of gestation. This review addresses similarities and differences among these species with respect to early embryonic development, hatching of the blastocyst from the zonal pellucida, pregnancy recognition signaling, and implantation of the conceptus to the uterine luminal epithelium.

**Abstract:**

The fertilization of oocytes ovulated by pigs, sheep, cows, and horses is not considered a limiting factor in successful establishment of pregnancy. Pig, sheep, and cow embryos undergo cleavage to the blastocyst stage, hatch from the zona pellucida, and undergo central-type implantation. Hatched blastocysts of pigs, sheep, and cows transition from tubular to long filamentous forms to establish surface area for exchange of nutrients and gases with the uterus. The equine blastocyst, surrounded by external membranes, does not elongate but migrates throughout the uterine lumen before attaching to the uterine luminal epithelium (LE) to begin implantation. Pregnancy recognition signaling in pigs requires the trophectoderm to express interleukin 1 beta, estrogens, prostaglandin E2, and interferon gamma. Sheep and cow conceptus trophectoderm expresses interferon tau that induces interferon regulatory factor 2 that inhibits transcription of estrogen and oxytocin receptors by uterine epithelia. This prevents oxytocin-induced luteolytic pulses of prostaglandin F2-alpha from regressing the corpora lutea, as well as ensuring the secretion of progesterone required for maintenance of pregnancy. The pregnancy recognition signal produced by equine blastocysts is not known. Implantation in these species requires interactions between extracellular matrix (ECM) proteins and integrins as the conceptus undergoes apposition and firm attachment to the uterine LE. This review provides details with respect to early embryonic development and the transition from spherical to filamentous conceptuses in pigs, sheep, and cows, as well as pre-implantation development of equine blastocysts and implantation of the conceptuses.

## 1. Introduction

A successful pregnancy is dependent on the quality of oocytes and sperm for fertilization and formation of a zygote that develops through the cleavage stages of embryonic development to the blastocyst stage. The blastocyst undergoes further development to a conceptus (embryo and its associate extra-embryonic membranes) in preparation for implantation and placentation. Mating females of agriculturally important species, including pigs, sheep, cows, goats, and horses, to males of proven fertility results in oocyte fertilization rates of 95 percent or greater [1]. After fertilization, embryos reside near the ampullary–isthmic junction of the oviduct before entering the uterus: 72 to 96 h in cows, 72 h in ewes, 48–56 h in pigs, and 144 h in mares [2]. Embryos of pigs [3,4] and sheep [5] retained in the oviduct fail to develop beyond the early blastocyst stage due to either a lack of essential factors for development or the presence of a factor(s) that inhibits development.

Zygotes of all mammalian species undergo cleavage to the morula stage, and then dividing cells are designated to contribute to either the inner cell mass (ICM), also called the embryonic disc, or to the trophectoderm, forming a blastocyst which must “hatch” out of the zona pellucida and expand to a large spherical blastocyst. Blastocysts then either elongate to a filamentous form, as in ruminants and pigs, or expand further in a spherical form until implantation is initiated, as in the mare. Estrogen (E2) and progesterone (P4) act via their receptors, ESR1 and PGR, respectively, in uterine epithelial and stromal cells to prepare the uterus for pregnancy, but downregulation of those receptors in the uterine epithelial cells is a prerequisite for implantation of the blastocyst/conceptus [6,7]. However, downregulation of PGR does not occur in endometrial stromal cells. Rather, P4 acts on uterine stromal cells to induce expression of growth factors, specifically fibroblast growth factors 7 (FGF7) and FGF10, which activate FGF receptor 2IIIB (FGFR2IIIB), a receptor tyrosine kinase, and hepatocyte growth factor (HGF), which activates its receptor encoded by cMet, also a receptor tyrosine kinase [8]. These growth factors have similar effects of stimulating proliferation, growth, migration, motility, and differentiation of cells. With downregulation of PGR in uterine epithelia, FGF7, FGF10, and/or HGF from stromal cells act in a paracrine manner to exert their respective effects on uterine epithelial cells and the trophectoderm, all of which express both FGR2IIIB and cMET [9,10,11,12]. The pig is an exception among livestock species in that the uterine epithelial cells express FGF7, but not FGF10 or HGF.

Estrogen from ovarian follicles and P4 from the corpus luteum (CL) prepare the uterus for production of prostaglandin F_2α_ (PGF) to induce regression of the CL and inhibit its production of P4 (luteolytic effect) if the female is not pregnant. Therefore, the conceptus must signal for maternal recognition of pregnancy to modify the pattern of secretion of PGF (ruminants and mare) or modify the direction of secretion from an endocrine direction in cyclic pigs to an exocrine direction into the uterine lumen for further metabolism (pigs). The pregnancy recognition signals interferon tau (ruminants) and E2 (pigs) will be discussed in detail later in this review. The pregnancy recognition signaling mechanism from conceptuses of mares has not been discovered.

The term “implantation” is used to describe the processes of attachment and invasion of the blastocyst into the endometrium of eutherian mammals. Implantation is a troublesome scientific term regarding pregnancy because it implies embedding of the blastocyst into the wall of the endometrium. Implantation is an appropriate term to use for humans, mice, and rats, in which the trophectoderm cells of the blastocysts breech the uterine LE and invade the endometrial stroma to provide a nest within the endometrium, in which the blastocyst further develops. However, the agriculturally important species, including pigs, sheep, cattle, and horses, exhibit central implantation, in which the blastocyst does not embed into the wall of the endometrium and instead develops within a placenta that is attached to the surface of the endometrium within the uterine lumen [13]. In these species, the placental trophectoderm cells either do not invade the endometrial LE, as is the case for pigs and (for the most part) horses, or there is limited invasion of trophectoderm cells into the uterine LE and fusion of trophectoderm cells with the uterine LE to form syncytial cells or plaques, as is the case for sheep and cattle [14,15]. Each of these species share the “Adhesion Cascade for Implantation”. The phases of this adhesion cascade include (1) shedding of the zona pellucida, (2) elongation of the trophectoderm and extraembryonic endoderm in pigs, sheep, and cattle, but not humans, mice, rats, and horses, (3) pre-contact and orientation of placental trophectoderm to the uterine LE, (4) apposition of the trophectoderm to the LE, and (5) adhesion of the apical surface of the trophectoderm to the apical surface of the LE. Attachment of trophectoderm to the uterine LE first requires removal of mucins from the glycocalyx of the uterine LE that sterically inhibit adhesion. Mucin removal allows for direct physical interactions between a mosaic of carbohydrates, glycans, and lectins at the apical surfaces of the opposing trophectoderm and uterine LE, which contribute to the initial attachment of the trophectoderm to the LE. These low-affinity contacts are then strengthened by a repertoire of adhesions between membrane-bound integrins and their ECM ligands, which provide stable adhesion of the trophectoderm to the uterine LE for implantation (see Figure 1) [16,17,18,19,20,21,22,23]. Implantation is the beginning of placentation, and placentation differs significantly among species [24,25,26].

## 2. Pregnancy in Pigs

### 2.1. General

Mating occurs during estrus that lasts 24 to 72 h, and the period of gestation is 114 days for pigs; see [6]. Functional CL producing P4 are required for the entirety of gestation, as the placenta produces little or no P4. Pig embryos are transported from the oviduct into the uterus around the four-cell stage between 60 and 72 h after the onset of estrus. The four-cell embryo continues through cleavage stages to reach the blastocyst stage on Day 5. Blastocysts, at 0.5 to 1 mm diameter, hatch from the zona pellucida between Days 6 and 7 and expand to 2 to 6 mm in diameter on Day 10 while migrating and spacing themselves throughout both uterine horns before elongating. The blastocysts elongate rapidly to a filamentous form of 100 to 200 mm in length between Days 11 and 12, and then they continue to elongate to 800 to 1000 mm in length by Day 16 of gestation. The initial period of elongation occurs at 30 to 45 mm/h, due mostly to cellular remodeling and hypertrophy, but further development to 800 to 1000 mm in length is due to both cell proliferation and migration. At least two conceptuses per uterine horn are required for maintenance of pregnancy in pigs.

Cellular remodeling occurs during the transition from spherical blastocysts to filamentous conceptuses; see [6]. The inner cell mass/embryonic disc (ICM) emerges through the covering trophectoderm (Rauber’s layer) when blastocysts are 2 to 5 mm in diameter. Thus, cells in the ICM have direct exposure to molecules present in the uterine lumen that are either secreted by or transported by the uterine LE and glandular epithelium (GE). Blastocysts greater than 5 mm in diameter have an ICM with short microvilli compared to long microvilli on trophectoderm cells. Trophectoderm cells of spherical blastocysts are dome-shaped and individually recognizable, whereas cells are flat in tubular blastocysts and not easily distinguishable from one another due to an increase in density of microvilli. Microvilli continue to be well developed in filamentous conceptuses except for the bulbous ends, which have few microvilli. The extra-embryonic endoderm cells of spherical blastocysts are spaced evenly with distinct nuclei and cell borders, but they later become indistinguishable except for being outlined by a thin line of short surface microvilli. Changes in the extra-embryonic endoderm in tubular blastocysts reflect changes in cellular reorganization as endodermal cells outside the elongation zone migrate via cellular extensions known as filapodia allowing them to be interconnected. However, endodermal cells in the elongation zone are densely packed, having a smaller diameter and more surface microvilli than those outside the elongation zone. The dense streak of endodermal cells in the elongation zone extends from the embryonic disc to the distal ends of the tubular blastocysts. At the edges of the elongation zone, many cells are migrating through extension of filopodia.

Total DNA and RNA in porcine conceptuses increase from Day 10 to 16, as do mitotic indices up to the 10–20 mm tubular stage of development, and then decrease about 40% with the onset of blastocyst elongation. Within the elongation zone, alterations in microfilaments and junctional complexes of trophectoderm cells and extension of filopodia from extra-embryonic endodermal cells allow movement and redistribution of cells toward the polar ends of tubular blastocysts. The actin cytoskeleton of pig trophectoderm cells between spherical to tubular and filamentous forms initially exhibits a pericellular distribution and then exhibits continuous actin-rich lateral borders and stress fibers along the basal surface [27,28,29,30,31]. The actin cytoskeleton, along with myosin II, generates force for conceptus elongation, as constricted regions along the length of filamentous conceptuses contain polarized trophectoderm cells with a distinct F-actin array. The orientation of microfilaments within the trophectoderm changes from horizontal to parallel relative to the lateral cell borders due to a complex cellular response to torsional forces generated by the elongation process and mediated through transmembrane integrin receptors and their associated focal adhesion complexes.

Evidence is mounting that the complex events of conceptus elongation in pigs are supported through the coordination of multiple metabolic biosynthetic pathways for glucose and fructose [6,14,32]. Oxidative metabolism primarily occurs through the TCA cycle and the electron transport chain, but proliferating and migrating cells, like the trophectoderm of pigs, enhance aerobic glycolysis. The glycolytic intermediates from glucose can then be shunted into the pentose phosphate pathway and one-carbon metabolism for the de novo synthesis of nucleotides. A result of aerobic glycolysis is limited availability of pyruvate for maintaining the TCA cycle, and trophectoderm cells likely replenish TCA cycle metabolites primarily through glutaminolysis to convert glutamine into TCA cycle intermediates. The synthesis of ATP, nucleotides, amino acids, and fatty acids through these biosynthetic pathways is essential to support elongation, migration, hormone synthesis, implantation, and early placental development of conceptuses.

### 2.2. Implantation

What is understood about the “Adhesion Cascade for Implantation” in pigs has been reviewed in detail [6,15]. Although the period of time when the trophectoderm of the pig blastocyst establishes firm attachment to the uterine LE, attachment that then supports subsequent tissue remodeling to fold the uterine placental interface, is reported to be somewhat different by various investigators, it is reasonable to define this period as being from Day 13 to Day 26 of pregnancy. By Day 26, placentation initiates by developing folds that increase the surface area and develop differentiated cells with cell type-specific expression of enzymes, receptors, and transporters necessary to optimize transport of water, ions, and nutrients across the uterine–placental interface to support fetal growth and development [30,31,32,33,34,35,36,37,38]. The electron micrographic studies of Dantzer [39] provide an excellent overview of the physical interactions between blastocysts and the uterus during this period. Both the trophectoderm of the blastocyst and uterine LE have a significant glycocalyx extending from their apical surfaces throughout implantation, but the glycocalyx of the uterine LE is always thicker than that of the trophectoderm. The apical surface of the uterine LE begins to physically protrude into the apical surface of trophectoderm cells on Days 13 and 14, serving to immobilize the blastocyst that was previously “free-floating” within the limited space of the uterine lumen. By Days 15 and 16 of pregnancy, microvilli form between the closely apposed apical plasma membranes of the trophectoderm and uterine LE cells. Between Days 15 and 20, the apical domes on the uterine LE develop cytoplasmic protrusions that extend between trophectoderm cells to the luminal space, and interdigitation of microvilli between the trophectoderm and uterine LE increases significantly through Day 26 of pregnancy [39] (See Figure 1).

In all eutherian mammals, temporary loss of mucin 1 (MUC1), a prominent component of the apical surface glycocalyx of the uterine LE with an extended carbohydrate configuration that physically inhibits attachment of the blastocyst trophectoderm to the uterine LE, is required before there can be firm attachment of the conceptus trophectoderm to the uterine LE [40]. Therefore, attachment of blastocyst trophectoderm to the uterine LE is predicated on P4 downregulating P4 receptors (PGR) in uterine LE [41,42], an event that is associated with downregulation in the expression of MUC1 on uterine LE [43]. The association between downregulation of PGR and MUC1 in uterine LE is strongly supported by the fact that MUC1 is lost at the apical surface of uterine LE of cyclic pigs given intramuscular injections of P4 to downregulate PGR [43].

Interactions between carbohydrates and lectins on the apical surfaces of the trophectoderm and uterine LE during blastocyst attachment for implantation have not been systematically investigated in pigs. However, loss of MUC1 from uterine LE is believed to expose low-affinity carbohydrate ligand binding molecules, including selectins, galectins, heparan sulfate proteoglycans, heparin binding EGF-like growth factors, cadherins, and CD44 that contribute to the initial attachment of blastocyst trophectoderm to uterine LE [19,44,45]. It is hypothesized that these carbohydrate ligands and their lectin receptors, expressed at the apical surfaces of the trophectoderm and uterine LE of pigs, mediate attach-and-release events between the respective cell types, similar to “Rolling and Tethering” of leukocytes to the endothelium for extravasation into connective tissues [46] and similar to events proposed for attachment of blastocysts to the uterine wall of humans [47]. Indeed, goats and sheep express H-type-1 antigens and glycosylation-dependent glycam-1, respectively, at the interface between the blastocyst trophectoderm and uterine LE during blastocyst adhesion for implantation [48,49].

These proposed low-affinity contacts between the apical surfaces of the blastocyst trophectoderm and uterine LE are stabilized by adhesion between a repertoire of integrins and their ECM ligands [22,31,50]. Integrin–ligand binding promotes cell–cell attachment, and integrins are major components of many adhesion cascades [51,52,53]. Integrins are transmembrane glycoprotein receptors composed of non-covalently linked α and β subunits, and the known 18 α- and 8 β-subunits can dimerize to form 24 heterodimer combinations that bind numerous ECM proteins [51,54,55]. In pigs, α1, α3, α4, α5, αv, β1, β3, and β5 integrin subunits are expressed at the apical surface of both the blastocyst trophectoderm and uterine LE, and P4 upregulates α4-, α5-, and β1-subunits on uterine LE during the peri-implantation period [43]. The integrin subunits detected at the apical surface of the uterine LE and trophectoderm of pigs potentially assemble into the αvβ1, αvβ3, αvβ5, α4β1, and α5β1 integrin receptors, and they have the opportunity to interact with ECM proteins expressed at the uterine–placental interface of pigs, including (1) fibronectin (FN1) [43], which binds αvβ3, α4β1, and α5β1, (2) vitronectin (VTN) [43], which binds αvβ3, (3) the latency-associated peptide (LAP) of transforming growth factor beta (TGFβ) [56], which binds αvβ1 and αvβ3, (4) the inter-α-trypsin inhibitor heavy chain-like protein (IαIH4) [57], which binds αvβ3, and/or (5) osteopontin (OPN, secreted phosphoprotein 1 (SPP1)) [58], which is upregulated by E2 [59] and binds αvβ1, αvβ3, αvβ5, α4β1, and α5β1. Affinity chromatography followed by immunoprecipitation have demonstrated direct binding of specific integrin receptors to ECM ligands on cultured uterine epithelial (pUE) and trophectoderm (pTr2) cells of pigs [60,61]. The integrin subunits αv, β1, β3, β5, β6, and β8 bind to LAP [61], whereas the αvβ6 integrin receptor on pTr2 cells and αvβ3 integrin receptor on pUE cells bind to OPN [60]. Integrin binding to OPN promotes dose-dependent attachment of pTr2 and porcine uterine epithelial cells and stimulates haptotactic cell migration directionally along a physical gradient of non-soluble OPN [60]. Knockdown of the αv-subunit in pTr2 cells by siRNA reduces pTr2 attachment to OPN and FN1 [62], and the αv-subunit co-localizes with talin-1 (TLN1) in integrin adhesion complexes (IACs) generated in the apical domain of pTr2 cells around OPN-coated microspheres cultured at the top of the cells [60,63].

Focal adhesions include transmembrane integrin heterodimer receptors (e.g., αvβ3, α5β1) that transmit diverse signals among proteins in the ECM, such as OPN and the actin cytoskeleton, to regulate growth, proliferation, survival, migration, gene expression, and changes in morphology of cells [29]. Integrins and associated proteins are “tensegrity structures” with molecular connections to the ECM, cytoskeletal filaments, and nuclear scaffolds for the transfer of mechanical signals through cells that induce integrated changes in cellular and nuclear structure (i.e., tension sensing) [64,65]. The resulting cytoskeletal complex or “IAC” physically links integrins to the ends of contractile microfilament bundle “stress fibers” to form a molecular bridge between the ECM and the cytoskeleton. Integrin adhesion complexes increase in size as tension increases across transmembrane receptors for integrins [66]. Applying force to the ECM affects integrins and associated IAC proteins that modify the shape of molecular complexes and elicit biochemical signals (mechanosensation) which change intracellular metabolism and gene expression (mechanotransduction). Elongation of pig conceptuses likely involves pathways, whereby serine–threonine kinases such as insulin growth factor 2 (IGF2) acting via receptor tyrosine kinases, integrin heterodimer-ECM complexes (e.g., OPN-αvβ3 and/or OPN-α5β1), and arginine act independently and in concert to activate mechanistic targets of rapamycin 1 (mTORC1) and/or mTORC2 cell signaling to induce proliferation, migration, cytoskeletal reorganization, and adhesion of trophectoderm cells to the uterine LE. The convergence of cell signaling pathways activated by IGF2, arginine, and OPN affect cytoskeletal proteins, including microfilaments (beta and gamma actin) and alpha actinin (bundles of actin that link actin cytoskeleton to the plasma membrane), intermediate filaments (cytokeratin, a major epithelial intermediate filament), and microtubules (beta tubulin), as well as myosin II, paxillin, parvins, ILK, and PINCH1, which are responsible for elongation of the porcine conceptus during the peri-implantation period of pregnancy.

Elongation of pig conceptuses occurs before initial attachment of the trophectoderm to the uterine LE to establish a non-invasive “central type” implantation [67]. Implantation of pig conceptuses is initiated as uterine LE and trophectoderm cells depolarize and initiate non-adhesive or pre-contact, apposition, and adhesion phases preceding establishment of an epitheliochorial placenta [45,68]. Attachment of the trophectoderm to the uterine LE requires loss of anti-adhesive molecules, mostly mucins, that inhibit attachment of the trophectoderm to the uterine LE [20,43,63]. This allows selectins and galectins [69] to facilitate initial attachment, followed by stable adhesive interactions between integrins and the maternal ECM—for example, OPN, a dominant contributor to implantation in pigs [22,46,63]. OPN, a secreted ECM protein that binds cell surface integrins, is expressed in increasing amounts by the uterine LE during the peri-implantation period of pregnancy in pigs [58,59]. Secretion of OPN is in response to E2 secreted by the conceptuses in discrete regions of the uterine LE juxtaposed to the conceptus trophectoderm on Day 13 and the entire uterine LE by Day 20, when there is stable adhesion of the conceptus trophectoderm to the uterine LE [58,59]. OPN is abundant at the utero-placental interface throughout pregnancy in pigs [58,59], as are integrin heterodimers that bind OPN [22,70].

### 2.3. Pregnancy Recognition Signaling (Figure 2)

Perry et al. [71] established that blastocysts and elongated conceptuses of pigs synthesize estrone (E1) and E2 from androstenedione, dehydroepiandrosterone, and P4 via aromatase, 17–20 desmolase, and sulphatase. Fischer et al. [72] confirmed and extended those results by co-culturing endometrial and conceptus tissues from Days 10.5, 11, 12, 16, and 25 of pregnancy and Day 25 of pseudopregnancy in Minimal Essential Medium containing [^3^H] P4. Spherical conceptuses did not produce estrone (E1) or estradiol (E2), but tubular and filamentous conceptuses produced both E1 and E2, and conceptus tissues from Days 16 and 25 (chorion) produced E1 (123 and 520 pg/mg tissue, respectively) and E2 (277 and 876 pg/mg tissue, respectively). Pig conceptuses synthesize and secrete E2 in biphasic peaks: one coincident with rapid conceptus elongation (Days 11–12) and another during the attachment phase (Days 15–18) of implantation [73,74]. Similarly, administration of exogenous estradiol to gilts from Days 11 to 15 of the estrous cycle extends CL maintenance and function for more than 60 days [75]. The duration of pseudopregnancy in cyclic gilts was maintained for only 25 days following administration of E2 either on Day 11 or from Days 14 to 18, but when administered on Day 11 and from Days 14 to 18 after onset of estrus, the length of pseudopregnancy was greater than 60 days [76].

**Figure 2 animals-14-01882-f002:**
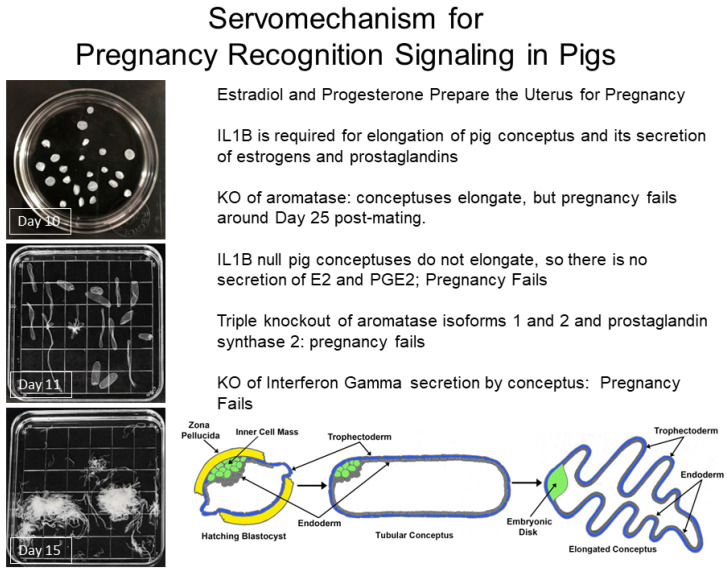
Pregnancy recognition signaling in pigs; see [77]. Interleukin 1-beta 2 is required for the elongation of pig conceptuses that is a prerequisite for synthesis and secretion of estradiol to induce exocrine secretion of prostaglandin F_2α_ into the uterine lumen and prostaglandin synthase 2 for production of prostaglandin E_2_ as a luteal protective hormone. The trophectoderm secretes interferon gamma to resolve inflammation to allow maintenance of pregnancy. KO is the abbreviation for gene knockout.

Research using gene editing has clarified the roles of sex steroids, prostaglandins, and cytokines in development of blastocysts and pregnancy recognition signaling in pigs; see [77]. Interleukin 1-beta 2 (IL1B2) is required for the elongation of pig conceptuses [78] that is a prerequisite for trophectoderm expression of isoforms of aromatase (CYP19A1, CYP19A2, and CYP19A3) for synthesis of E2 [79,80] and prostaglandin synthase 2 (PTGS2) for synthesis of PGE2. Pig conceptuses express CYP19A1 for initial production and secretion of E2 (Days 11–12), while CYP19A2 is the active aromatase between Days 15 and 30 of gestation. Only when both CYP19A1 and CYP10A2 were silenced did the trophectoderm express CYP19A3. E2 redirects secretion of PGF from an endocrine mode into the uterine vasculature (endocrine secretion) and the uterine lumen (exocrine secretion) [81], where it may be converted to PGE2 or to its inactive metabolite (13, 14, dihydro-15-keto-prostaglandin F; PGFM) by 15-prostaglandin dehydrogenase-delta^13^ reductase [82]. The E2-dependent exocrine secretion of PGF into the uterine lumen prevents pulsatile release of PGF into the uterine venous system that would otherwise induce CL regression and the lack of P4 required for establishment and maintenance of pregnancy.

A triple knockout of CYP19A1, CYP19A2, and PTSG2 in pig conceptuses was required to prevent secretion of estradiol and PGE2 for pregnancy recognition signaling and maintenance of the CL beyond Day 16 post-mating and normal development of conceptuses; see [83]. Only this triple knockout resulted in failure of pig blastocysts to fully elongate and establish pregnancy, whereas the double knockout of CYP19A1 and CYP19A2 in blastocysts allowed conceptuses to fully elongate before the pregnancies failed around Day 25 after onset of gestation. Thus, PGE2 appears to have a central role in maintenance of functional CL in pigs.

The endocrine–exocrine theory of pregnancy recognition in the pig is based on evidence for uterine release of PGF toward the vasculature (endocrine secretion) in cyclic gilts, as compared to uterine release of PGF into the uterine lumen (exocrine secretion) in pregnant gilts [81]. If PGE2 has a role in CL maintenance, how is it transported out of the uterus when PGF is not? Transporters of PGE2 in the uterus may favor transport into the systemic circulation and/or across the utero-ovarian plexus for delivery to the CL. First, the ratio between total recoverable PGE vs. PGF (ng) in uterine flushings is greater for pregnant than cyclic gilts on Days 10.5, 11, 11.5, 12, and 14 by 3:1, 5:1, 4:1, 3:1, and 2:1, respectively [74]. Seo et al. [84] reported that the uterine endometrium of pigs expresses mRNAs for ATP-binding cassette sub-family C member 4 (ABCC4, also known as multidrug-resistant protein 4, MRP4) and solute carrier organic anion transporter family member 2A1 (SLCO2A1, also known as prostaglandin transporter, PGT), with the greatest expression on Day 12 of pregnancy and during late pregnancy. Expression of ABCC4 mRNA and protein was localized to the uterine LE and GE, whereas expression of SLCO2A1 mRNA and protein was localized to the uterine LE and endothelial cells of blood vessels. SLCO2A1 supports influx or efflux of prostaglandins (PGs) depending on concentrations of lactate but preferentially promotes uptake of PGs from the extracellular space. However, Russel et al. [85] and Lin et al. [86] reported that ABCC4 transported PGE1 and PGE2 with the highest affinity and knocking out the ABCC4 gene decreased PGEs in plasma of mice compared to that for wild-type mice. Further, Rius et al. [87] reported that ABCC4 transported PGE2 away from epithelial cells of seminal vesicles with higher affinity values (Km) for PGE2 (3.5 µM) than thromboxane B2 (9.9 µM) and PGF (12.6 µM). Expression of these two transporters of PGs in endometrial explants from pigs increases in response to IL1B and may account for transport of both PGE2 and PGF, but it favors preferential transport PGE in an endocrine direction in pregnant pigs to maintain a functional CL.

Pig conceptuses produce interferons between Days 12 and 20 of pregnancy [88] that include interferon delta (IFND), a type I interferon, and interferon gamma (IFNG), a type II interferon [89], with IFNG being the predominant interferon in the pig [90]. There is no evidence that IFND or IFNG are pregnancy recognition signals to prevent luteolysis; however, intrauterine infusion of Day 15 pig conceptus secretory proteins does increase concentrations of PGE2 and PGFM in plasma from the inferior vena cava, and PGE2 may enhance maintenance of the structural and functional integrity of the CL [91]. IFND and IFNG from porcine conceptuses also have paracrine effects via their receptors on cells of the uterus to regulate expression of interferon-stimulated genes (ISGs) involved in complex communication between maternal and conceptus systems [46]. In the case of IFNG, the conceptus trophectoderm releases extracellular vesicles containing IFNG that are taken up by the uterine LE and result in the induction of ISGs in the uterine stroma and recruitment of T cells to sites of conceptus implantation [92,93]. D’Andrea et al. [94] detected local effects of conceptus IFNs on endometrial cells and the trophoblast by demonstrating induction of 2′,5′ oligoadenylate synthetase activity in the trophoblast after IFN treatment, as well as in endometrial epithelial and stromal cells. No autocrine effects of conceptus IFNs on the pig trophoblast were detected, as those cells lacked detectable receptors for either IFNG or IFND [94]. However, the same group later detected IFNGR1 mRNA by RT-PCR in uterine epithelial and stromal cells, as well as embryonic tissues, from as early as Day 10 of pregnancy. Expression of IFNGR1 in the trophoblast appeared to be developmentally regulated, with weak expression on Days 12 and 15 of gestation similar to that for some IFNG-sensitive cells on Day 16 of gestation [95].

IFNG may influence conceptus attachment through effects on integrins and heparin sulphate proteoglycans [96], but neither IFNG nor IFND seem to affect remodeling of uterine epithelia, cellular polarity, or receptivity to trophoblast attachment [94]. Pig conceptus IFNs increase expression of many endometrial genes, including class I and II major histocompatibility complexes, STAT1, and IRF1, involved in immune regulation [97,98,99]. Additionally, pig conceptus IFNs induce endometrial expression of fas-ligand (FASLG), tumor necrosis factor 10 (TNFSF10), and TNFSF receptor (FAS) on Day 15 of pregnancy [100,101]. It has also been reported that IFNG increases expression of cysteine-X-cysteine chemokine ligands (CXCL) 9, 10, 11, and 12, which stimulate migration of T-cells and NK cells, which is consistent with effects of interferons that recruit immune cells into the endometrium during the implantation period of pregnancy in pigs [100,101].

To understand the role of IFNG in the pregnant uterus, IFNG null conceptuses were transferred into the uteri of recipient gilts [102]. The result was hyperinflammation of the uterus and failed pregnancies. Thus, the role of IFNG appears to be to resolve inflammation through an unidentified mechanism to allow a state of tolerance required for the uterus to be receptive to implantation, placentation, and a successful pregnancy. Cooperative effects of E2 with IFNG may also induce expression of interferon regulatory factor 2 (IFR2), a repressor of gene transcription, in the uterine LE to restrict expression of classical ISGs to uterine GE and stromal cells [98]. P4 is permissive to the establishment of a uterine environment that supports development of pig conceptuses throughout gestation [103].

### 2.4. Uterine Histotroph

Uterine fluids of pigs contain many molecules secreted or transported into the uterine lumen to form histotroph [104]. Histotroph includes uteroferrin, now known as ACP5 (phosphatase, acid, type 5, tartrate resistant), which transports iron to the conceptus for erythropoiesis and stimulation of hematopoiesis [42]. Other components of histotroph include retinol binding protein, plasmin/trypsin inhibitor, leucine aminopeptidase, glucose phosphate isomerase, serine protease inhibitors, lysozyme, various proteases, hexosaminadase, phospholipases, prostaglandin synthases, insulin-like growth factors 1 and 2, insulin-like growth factor binding proteins, high-molecular weight glycoproteins, glucose, fructose, ascorbic acid, amino acids, prostaglandins, cyclic nucleotides, catecholamines, calcium, sodium, potassium, E2, P4, and colony-stimulating factor 1. Also included are OPN, integrins, mucins, FGF7, FGF10, HGF, staniocalcins, transforming growth factors beta-1, -2, and -3, transporters for amino acids and glucose, nitric oxide synthases, GTP cyclohydrolase 1, tetrahydrobiopterin, interleukins 1 and 4, Mx, ISG15, beta 2 microglobulin, histocompatibility antigens, interferon regulatory factors 1, 2, and 6, 2′,5′-oligoadenylate synthetase, galectins, cystatins, cathepsins, Wnt gene products, uterocalin, uteroglobin, gastrin-releasing peptide, hypoxia-inducible factor isoforms, mechanistic target of rapamycin (MTOR), ornithine decarboxylase (ODC1), and polyamines. There is much to be learned about the roles of these molecules, which are secreted and/or transported into the uterine lumen by uterine epithelial and stromal cells to create a uterine microenvironment that supports conceptus development. The uterine GE is required for conceptus development and successful establishment and maintenance of pregnancy in pigs, sheep [105,106], and humans [25].

The growth factors FGF7 and FGFR2IIIB are expressed by endometrial epithelia of gilts, and expression is greatest between Days 12 and 15 of the estrous cycle and pregnancy [11,12]. FGF7 is also present in uterine secretions on Day 12 of the estrous cycle and pregnancy, and FGFR2IIIB is expressed by the conceptus trophectoderm. Thus, FGF7 may act on uterine epithelial cells in an autocrine manner and on the conceptus trophectoderm in a paracrine manner to stimulate proliferation, differentiation, and migration of cells, as well as angiogenesis. The pig uterus is unique in that FGF7 is expressed by uterine LE and FGFR2IIIB is expressed by the uterine LE and GE, as well as the trophectoderm, to affect proliferation and differentiated cell functions during conceptus development and implantation. FGF7 expression by uterine LE of pigs increases between Days 9 and 12 of the estrous cycle and pregnancy, as circulating concentrations of P4 increase and PGR decrease in uterine epithelia, while E2 from conceptuses increases to transactivate ESR1 expressed by uterine LE. Progesterone is permissive to FGF7 expression by downregulating PGR in the uterine LE, which allows E2 to induce expression of FGF7 that influences growth and development of porcine conceptuses [11]. Neither FGF10 nor HGF are detectable in uterine stromal cells during the peri-implantation stage of pregnancy in pigs.

### 2.5. Interferon-Stimulated Genes

Pig conceptuses secrete E2 as a component of the pregnancy recognition signaling molecules. Joyce et al. [46,97,98,99] evaluated expression of interferon regulatory factor 1 (IRF1), an inducer of gene transcription, and IRF2, an inhibitor of gene transcription, in uteri of pregnant gilts. Both IRF1 and IRF2 were detectable in uteri during the estrous cycle; however, expression of classical ISGs, including IRF1, signal transducer and activator of gene transcription 2 (STAT2), MHC, and beta-2 microglobulin (B2M), increased on Day 12 of pregnancy, but only in uterine GE and stromal cells. IRF2 increased in uterine LE after Day 12 of pregnancy, and E2 increased its expression to restrict expression of IRF1 and other classical ISGs in response to conceptus-derived IFND and/or IFNG to uterine GE and stromal cells of the uterus. Inhibition of expression of classical ISGs, swine leukocyte antigens, and B2M by uterine LE is due to E2-induced expression of IRF2. Thus, an immunologically favorable environment at the trophectoderm–uterine LE interface favors survival of the conceptus allograft.

## 3. Pregnancy in Sheep

### 3.1. General

The zygote undergoes cleavage stages in the oviduct of sheep before entering the uterus at the morula stage on Day 3 or Day 4 and reaching the blastocyst stage by Day 6 of gestation. The blastocyst hatches from the zona pellucida between Days 8 and 9, and pluripotent blastomeres differentiate into the ICM and trophectoderm to form the embryo (ectoderm, mesoderm, and endoderm) and trophectoderm, respectively. The blastocyst hatches from the zona pellucida between Days 8 and 9 (200 µm in diameter and about 300 cells) and increases in diameter (400–900 µm in diameter and 400–900 cells) before elongating. The spherical conceptus transitions to a tubular form by Day 11, and then into a long filamentous form between Days 12 (10–22 mm), 14 (10 cm), and 17 (25 cm) of gestation. During elongation, the conceptus is unattached to the uterine LE and dependent on the nutrients secreted and/or transported into the uterine lumen that make up histotroph. Conceptus elongation greatly increases the surface area of trophectoderm attached directly to the uterine LE, availability for absorption of components of histotroph, and pregnancy recognition signaling for establishment of pregnancy; see [14].

### 3.2. Uterine Histotroph

Sheep conceptuses elongate while exposed to histotroph from the uterine glands that contains hormones, enzymes, growth factors, cytokines, transport proteins, adhesion factors, nutrients, and other substances that are required for conceptus development, implantation, and placentation. The MTOR/FRAP1 cell signaling pathway regulates cell growth and metabolism, as it is a “nutrient sensing system” stimulated by molecules that include secreted OPN, glucose and fructose, and amino acids, including arginine; see [107]. The FK506-binding protein 12-rapamycin-associated protein 1 (FRAP1), target of rapamycin complex subunit LST8 (LST8), target of rapamycin complex 2 subunit MAPKAP1 (MAPKAP1), regulatory-associated protein of mTOR (RAPTOR), RPTOR independent companion of MTOR complex 2 (RICTOR), tuberous sclerosis 1 (TSC1), tuberous sclerosis 2 (TSC2), ras homolog enriched in brain (RHEB), and eukaryotic translation initiation factor 4E-binding protein 1 (EIF4EBP1) are expressed by the endometrium and conceptus trophectoderm of sheep between Days 13 and 18 of pregnancy. The increases in expression of RICTOR, RHEB, EIF4EBP1, and RHEB are associated with rapid growth and development of ovine conceptuses [108]. P4 and interferon tau (IFNT, the pregnancy recognition signal in sheep) stimulate expression of RHEB and EIF4EBP1 in the endometrium of sheep [108], while MTORC1 is localized to the cytoplasm, and phosphorylated MTOR is localized to nuclei of ovine trophectoderm cells [109].

### 3.3. Pregnancy Recognition Signaling (Figure 3)

The mononuclear trophectoderm cells synthesize and secrete IFNT between Days 10 and 21 of gestation as the signal for maternal recognition of pregnancy in sheep and other ruminants; see [110]. IFNT acts via its receptors, IFNAR1 and IFNAR2, on uterine LE, sGE, and GE to induce expression of IRF2, a repressor of gene transcription that silences transcription of ESR1, which prevents E2-induced expression of oxytocin receptors (OXTRs). Thus, pulsatile release of oxytocin (OXY) from CL and the posterior pituitary is unable to induce the luteolytic pulses of PGF required to regress the CL. Maintenance of secretion of P4 by CL is required for successful pregnancies in sheep and other mammals. During the period of pregnancy recognition signaling, P4 auto-downregulates expression of PGR in uterine epithelia as a prerequisite for the P4-induced secretion of histotroph required for growth and development of the conceptus and implantation. Although P4 is required for secretion and/or transport of components of histotroph, PGRs are not expressed by uterine epithelia. Rather, P4 mediates its effects in a paracrine fashion via PGR-positive stromal cells that express growth factors, known as progestamedins, including FGF7, FGF10, and HGF [9,10]. HGF transactivates a receptor encoded by c-met and localizes to uterine epithelia, the trophectoderm, and the chorioallantoic mesenchyme. HGF stimulates morphogenesis and differentiation of cells required for establishment and maintenance of pregnancy, conceptus implantation, and placentation. FGF10 mRNA is expressed by endometrial stromal cells and mesenchymal cells of the chorioallantoic placenta while FGF7mRNA is localized to the tunica muscularis of blood vessels in the endometrium and myometrium. Fibroblast growth factor receptor 2IIIb (FGFR2IIIb) is the high-affinity receptor for both FGF10 and FGF7 and is expressed by uterine epithelia and the conceptus trophectoderm. Thus, FGF10 is a stromal cell-derived mediator of uterine epithelial and conceptus trophectodermal functions similar to those of HGF, while FGF7 likely influences vascular development in the uterus and conceptus.

**Figure 3 animals-14-01882-f003:**
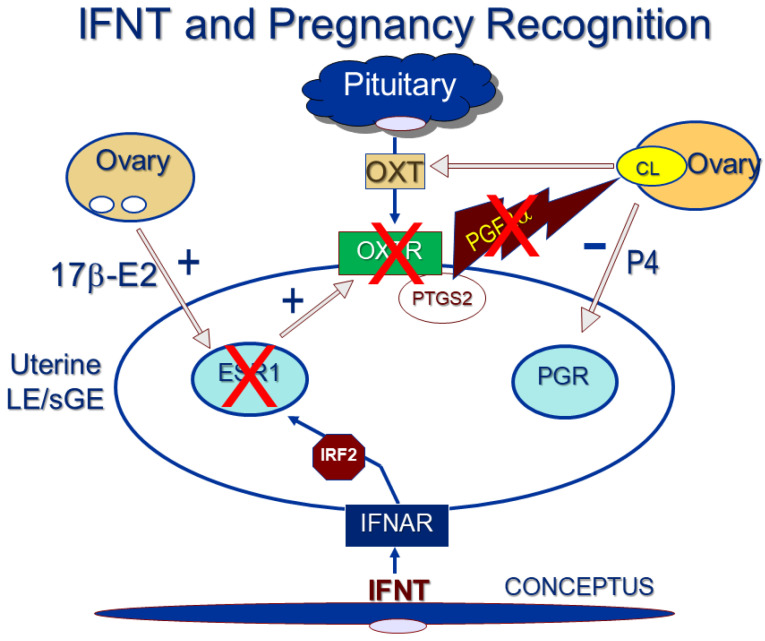
The model for pregnancy recognition signaling in sheep, cows, and goats; see [106]. Interferon tau (IFNT) secreted by the conceptus trophectoderm between Days 10 and 21 of gestation binds Type I IFN receptors (IFNAR) on uterine epithelia to induce interferon regulatory factor 2 (IRF2) that is a potent inhibitor of transcription. IRF2 inhibits expression of receptors for estradiol (ESR1) and oxytocin (OXTR), preventing estradiol-induced expression of OXTR and release of oxytocin (OXY)-induced luteolytic pulses of prostaglandin F_2α_ to allow maintenance of corpora lutea to produce the progesterone required for the establishment and maintenance of pregnancy.

IFNT increases expression of ISGs in uterine stroma and GE of sheep, with P4 being permissive to its effects [111,112,113]. Temporal and cell-specific expression of ISGs by uterine GE and stromal cells varies, but most have the same pattern of expression as interferon-stimulated gene 15 (ISG15) [111]. ISG15 is expressed by the uterine LE and stratum compactum stroma (SCS) from Day 13 of pregnancy, prior to implantation, and expression extends to the stratum spongiosum stroma (SSS) by Day 15, when implantation is underway. Expression of ISG15 is maintained throughout the stroma through Day 25 and declines by Day 30 of pregnancy, but only in some areas of the maternal–conceptus interface [114]. Classical ISGs like ISG15 are not expressed by the uterine LE and sGE during pregnancy, as IFNT-induced expression of IRF2 in the uterine LE and sGE [115] suppresses transcription of those genes [116]. The uterine LE and sGE do not express MHC1 and B2M, which may prevent immune rejection of the semi-allogeneic conceptus [116]. Relevant to pregnancy recognition signaling, Arosh et al. [117] reported that luteal cells of nonpregnant sheep produce PGF during luteolysis but produce PGE2 during pregnancy in response to IFNT. The authors suggest that PGE2 from the uterus is transferred across the utero-ovarian vascular plexus for maintenance of CL and their secretion of P4.

The roles of classical ISGs in the ovine uterus are not well defined. However, ISG15 is involved in conjugation of intracellular proteins that undergo degradation in the proteasome [118], or ISG15 may stabilize proteins for modification [119]. ISG15 forms stable conjugates with endometrial proteins of sheep and cows and may target proteins for pregnancy-associated regulation and/or modification [102,114].

OPN, a prominent component of the uterine environment during pregnancy in sheep, induces motility in trophectoderm cells through MTOR signaling, resulting in F-actin reorganization and phosphorylation of IAC proteins stimulated by IGF1 (insulin-like growth factor 1), including focal adhesion kinase (FAK) [50]. Crosstalk among multiple membrane and intracellular cellular signaling cascades activated by OPN includes MTOR and integrin binding to ovine trophectoderm cells to stimulate proliferation, migration, and attachment to uterine LE during the peri-implantation period of pregnancy [120]. Binding of OPN to αvβ3 and possibly α5β1 integrin heterodimers induces IAC assembly as a prerequisite for adhesion and migration of trophectoderm cells through activation of the following: (1) P70S6K via crosstalk between FRAP1/MTOR and MAPK pathways; (2) MTOR, PI3K, MAPK3/MAPK1 (ERK1/2), and MAPK14 (P38) signaling; and (3) focal adhesion assembly and myosin II motor activity [120].

Arginine, a major component of uterine histotroph in sheep, stimulates proliferation, migration, and protein synthesis in trophectoderm cells [121,122]. Arginine increases the following: (1) phosphorylation of RPS6K in a dose-dependent manner; (2) phosphorylation of RAC-alpha serine/threonine-protein kinase (AKT1), RPS6K, and RPS6; (3) nuclear phosphorylated RPS6K and cytoplasmic phosphorylated RPS6; and (4) proliferation and migration of trophectoderm cells [121]. Phosphorylation of RPS6K and RPS6 can be blocked by inhibitors of both PI3K and MTOR cell signaling, and L-arginine, but not D-arginine, activates MTOR cell signaling via phosphorylation of RPS6K and RPS6 [120]. Arginine also stimulates cell proliferation after being metabolized to nitric oxide (NO) via NO synthase 1/2 (NOS1/NOS2) or via arginase to ornithine, which is converted to polyamines by ornithine decarboxlylase 1 (ODC1) [122]. The polyamines include putrescine, spermidine, and spermine [122]. Two NO donors, S-nitroso-N-acetyl-DL-penicillamine (SNAP) and diethylenetriamine NONOate (DETA), increase proliferation of trophectoderm cells, as does putrescine. Both L-NAME (NOS inhibitor to reduce NO synthesis) and nor-NOHA (arginase inhibitor to block synthesis of putrescine) decrease proliferation of trophectoderm cells. Both NO and polyamines stimulate proliferation and migration of trophectoderm cells essential to elongation of sheep conceptuses, and supplemental dietary arginine enhances embryonic survival in sheep, pigs, mice, and rats [123,124].

Total recoverable glucose in the uterine lumen increases 12-fold between Days 10 and 15 of pregnancy, as does glucose transporter 1 (SLC2A1) in endometria of ovariectomized ewes in response to P4, and there is an additional 2.1-fold increase in response to IFNT. SLC2A3 is expressed by the conceptus trophectoderm [125]. Concentrations of fructose in uterine flushings of sheep have not been reported except for on Day 17 of gestation, when total recoverable fructose and glucose were 3.6 and 0.14 mg, respectively [126]. For ovine trophectoderm cells, (1) both fructose and glucose stimulate proliferation via the MTOR pathway; (2) phosphorylation of RPS6K and EIF4EBP1 in response to fructose requires both PI3K and MTOR and glutamine-fructose-6-phosphate transaminase 1 (GFPT1); and (3) inhibition of the hexosamine biosynthesis pathway by azaserine blocks MTOR-RPS6K and MTOR-EIF4EBP1 signaling and fructose stimulation of proliferation of oTr cells [127]. Thus, fructose and glucose support growth and development of sheep conceptuses as trophectoderm cells use the serinogenesis pathway for one-carbon metabolism for synthesis of (1) purines, required for the synthesis of nucleotides and nucleic acids; (2) thymidine, required for the synthesis of DNA; and (3) S-adenosylmethionine (SAM), the principal biological methylating agent for epigenetic modifications. Also, using explant cultures of Day 16 sheep conceptuses, glucose increased the total and phosphorylated forms of MTOR, ODC1, NOS2, and GTP cyclohydrolase 1 (GCH1); see [128].

Sheep conceptuses require nutrients primarily secreted and/or transported into the uterine lumen by the uterine GE to form histotroph, as described previously; see [128]. Pregnancies fail in ewes lacking a uterine GE, as conceptuses do not elongate or survive [129]. The uterine epithelial cells express transporters for amino acids [130,131], glucose [132,133], fructose [134,135], polyamines [133], minerals [136,137], and creatine [138]. Glucose and fructose are transported via solute carrier family 2A (SLC2A8) or sodium-dependent glucose transporters (SLC5A1). The glucose transporters SLC2A1, SLC2A4, and SLC5A1 are expressed by the uterine LE and GE, as well as cells of the conceptus during the peri-implantation period of pregnancy [139]. Fructose is transported via SLC2A5, and SLC2A8 can transport both glucose and fructose [140,141]. During the peri-implantation period of pregnancy in sheep, SLC2A5 and SLC2A8 mRNAs are expressed by the conceptus [125]. After implantation and during placentome development, SLC2A8 localizes to the uterine and chorionic epithelia in both the cotyledonary and caruncular portions of placentomes [142].

### 3.4. Implantation

What is understood about the “Adhesion Cascade for Implantation” in sheep has been reviewed in detail [14,31,32,45]. The electron micrographic studies of Guillomot [18,143] provide an excellent overview of the physical interactions between the blastocyst and uterus during this period. By Day 14, the apical surface of elongating trophectoderm cells are closely apposed to the apical surfaces of the uterine LE, with the LE cells protruding into the trophectoderm. At this time, the blastocyst is immobilized within the uterine lumen but can still be recovered intact from the uterus by lavage with only superficial damage. Between Days 13 and 15, there is a reduction in microvilli on the trophectoderm, but not the uterine LE. By Day 16, apposition begins near the ICM and spreads towards the ends of the elongated blastocyst with development of cytoplasmic projections of trophectoderm cells and uterine LE microvilli that firmly adhere the blastocyst to the uterus so that uterine lavage to recover the blastocyst damages the tissues. Interdigitation of apical projections between the trophectoderm and uterine LE occurs in both caruncular and intercaruncular regions of the uterus and is completed by Day 22 of pregnancy [18,143]. Between Days 14 and 16 of gestation, 15–20% of the trophectoderm cells undergo consecutive nuclear divisions without cytokinesis to form binucleate trophoblast giant cells (TGCs) which express pregnancy-associated glycoproteins (PAGS) and fuse with uterine LE to form syncytia. The extent of and mechanistic details of development of these syncytia continue to be investigated [144,145,146,147].

The current consensus for the cascade of events underlying attachment of the blastocyst to the uterus in sheep begins with downregulation of MUC1 at the apical surface of uterine LE cells across both caruncular and intercaruncular regions of the uterus. Unlike in pigs, P4 does not appear to downregulate MUC1 expression on the uterine LE [43,63], but temporary loss of uterine LE may be responsible for a decrease in MUC1 at the uterine–placental interface in ruminants [148,149,150,151]. MUC1, a glycocalyx molecule, extends a substantial distance outward from the apical surface of the uterine LE, and its removal allows interactions between lectin and integrin receptors present on the apical surfaces of the closely apposed trophectoderm and uterine LE to form bridging ligands. Molecules that potentially mediate attachment of the trophectoderm to the uterine LE in sheep include glycosylation-dependent cell adhesion molecule 1 (GLYCAM-1), galectin 15 (LGALS15), OPN, and integrin receptors [49,151,152,153]. Initial attachment is likely mediated by GLYCAM1 and LGALS15, and firm attachment is likely mediated by OPN binding to integrin receptors [45,49,70,108,110,154,155]. LGALS15 expression is induced by P4 and is further increased by IFNT [152], and OPN is induced by P4 in the uterine GE and secreted into the uterine lumen [70,154]. The integrin subunits αv, α4, α5, β1, β3, and β5 are constitutively expressed on the apical surfaces of the trophectoderm and uterine LE, with the potential to form integrin receptors αvβ3, αvβ1, αvβ5, α4β1, and α5β1 during the peri-implantation period of pregnancy [63,151].

There is evidence for integrins influencing early pregnancy in sheep. First, morpholino antisense oligonucleotides were used to block the translation of mRNA for the β3 integrin subunit in the blastocyst trophectoderm, and although the blastocysts elongated and attached to the uterus, the embryos were smaller than normal on Day 25, and the allantois of the developing placenta had decreased levels of OPN and nitric oxide synthase 3 (NOS3), suggesting effects on development of the allantoic vasculature that transports nutrients to support growth and development of the embryo [155]. Second, the β5 subunit extends from the apical surface of the uterine LE into the sGE, where it can interact with trophectoderm papillae that actively uptake histotroph and likely act as tethers against which forces necessary to generate elongation are applied [150]. Third, integrin subunits αv, α4, β1, β3, and β5 are present at the apical surface of the trophectoderm and uterine LE from Day 11 to 16, implying roles for αvβ3, αvβ5, and α4β1 integrin receptors in attaching the trophectoderm to the uterine LE for implantation [63]. Affinity chromatography and immunoprecipitation experiments determined that the αvβ3 integrin receptor on ovine trophectoderm cells (oTr1) binds OPN [120]. Further, arginine-glycine-asparagine (RGD)-mediated interaction between integrins and OPN stimulate oTr1 cell adhesion, the αv integrin subunit incorporates into IACs around OPN-coated microbeads at the apical surface of oTr1 cells [120], integrins interact via the RGD sequence of OPN to activate mitogen-activated protein kinase (MAPK) p38 and p70 ribosomal protein S6 kinase beta-1 (P70S6K) [120], and IACs containing the β3 integrin subunit increase at the base of cultured oTr1 cells in response to treatment with the combination of arginine and OPN [156].

### 3.5. Progesterone and Elongation of Conceptuses

Treatment of ewes with 25 mg P4 in corn oil daily from 36 h after onset of estrus and breeding to either Day 9 or 12 of pregnancy accelerates development of conceptuses as compared to that for ewes treated daily with corn oil only [157,158]. Treatment of the ewes with P4 also increased expression of ISGs, including cathepsin L, radical S-adenosyl methionine domain containing 2, and galactin 15 [157]. In a subsequent study evaluating expression of FGF7, FGF10, HGF, IGF1, IGF2, MET, FGFR2IIIB, and IGFBPs mRNAs in the uterine endometrium, expression of FGF10 and MET mRNAs increased between Day 9 and Day 12 in association with accelerated development of blastocysts on Days 9 and 12 of pregnancy [159]. There was no effect of P4 treatment on endometrial expression of FGF7, IGF1, IGF2, and IGF1R mRNAs, but expression of IGFBP1 and IGFBP3 mRNAs in the uterine LE was greater for P4-treated ewes on Day 12 of pregnancy. FGF10, MET, IGFBP1, and IGFBP3 are P4-regulated factors within the ovine uterus associated with endometrial function and rate of development of blastocysts during the peri-implantation period of gestation [159]. For ewes treated with P4, only arginine and lysine were more abundant in uterine flushings from P4-treated ewes, and expression of transporters for glucose, SLC2A1 and SLC5A1, and basic amino acid SLC7A2B was greater in the uterine LE and sGE of P4-treated ewes on both Days 9 and 12 of gestation [160]. Early treatment of ewes with P4 also affected expression of the endometrial WNT system, specifically WNT2 in stromal cells and WNT11 and WNT7A in the uterine LE [161].

Treatment of pregnant ewes with 25 mg of P4/day from 36 h after onset of estrus and breeding to either Day 9 or Day 12 of pregnancy also resulted in temporal and cell-specific changes in endometrial expression of occludin, tight junction protein 2, and claudins 1–4 that transiently decreased tight junction-associated proteins, but tight junction-associated proteins increased between Days 14 and 16 of pregnancy [162]. Cadherin 1 and beta-catenin forming adherens junctions decreased after Day 10 of pregnancy in the uterine LE but increased by Day 16 with the onset of implantation. Perhaps P4 induces transient decreases in tight and adherens junctions in the uterine LE between Days 10 and 12 to increase selective serum and tissue fluid transudation that enhances blastocyst elongation for implantation [162].

A subset of the ewes in the study by Hoskins et al. [158] were allowed to go to Day 125 of gestation to determine if advancing the time of elongation of conceptuses on Days 9 and 12 affected fetal-placental development at Day 125 of gestation [133]. However, ewes treated with exogenous P4 had increased amounts of aspartate and arginine in allantoic fluid and amniotic fluid, respectively [133]. Ewes that received exogenous P4 also had greater expression of mRNAs for SLC7A1, SLC7A2, SLC2A1, AGMAT, and ODC1 in endometria, as well as SLC1A4, SLC2A5, SLC2A8 and ODC1 in placentomes. In addition, AZIN2 protein was immunolocalized to the uterine LE and GE in P4-treated ewes, whereas AZIN2 was localized only to the uterine LE in ewes treated with corn oil without P4 [133].

## 4. Pregnancy in Cows

### 4.1. General

Chang [163] reported that bovine blastocysts are spherical from Days 8 to 9 (0.17 mm diameter), oblong or tubular by Days 12 to 13 (1.5 to 3.3 mm by 0.9 to 1.7 mm), and filamentous from Days 13 to 14 (1.5 × 10 mm), 14 to 15 (2 × 18 mm), 16 to 17 (1.8 × 50 mm) and 17 to 18 (1.5 × 160 mm) of pregnancy. On Days 17 to 18 of pregnancy, bovine conceptuses occupy two-thirds of the gravid uterine horn. They occupy the entire gravid uterine horn by Days 18 to 20 and a portion of the contralateral uterine horn by Day 24. Shemesh et al. [164] reported that bovine blastocysts from Days 13, 15, and 16 of pregnancy produce P4, some testosterone, and limited quantities of E2. Eley et al. [165] also detected the conversion of androstenedione to E2 by bovine conceptuses from Days 15 to 17 of gestation and extensive metabolism of P4 and androstenedione to their 5α-reduced metabolites [166].

After fertilization of the oocyte in the oviduct, bovine embryos enter the uterus at about the 16-cell stage on Day 4 of pregnancy with development through the morula stage, reaching the blastocyst stage by Day 7 of pregnancy. Blastocysts hatch from the zona pellucida from Days 9 to 10, forming expanded blastocysts by Days 12 to 13, tubular conceptuses on Days 14 to 15, and filamentous conceptuses by Days 16 to 17 of gestation [167]. Between Days 12 and 17 of pregnancy, bovine blastocysts increase from 150 µm in diameter to 130 to 150 mm in length [168,169]. After Day 19, the fully elongated conceptus begins implantation with firm apposition and attachment of the trophectoderm to the uterine LE. During elongation, the bovine conceptus increases in size by some 1000-fold [170], associated with an increase in protein content [171,172].

### 4.2. Growth Factors and Embryonic Development

Watson et al. [173] used in vitro produced bovine embryos to evaluate patterns of gene expression from zygote to hatched blastocyst stages and identified transcripts for transforming growth factor alpha (TGFA) and platelet-derived growth factor (PDGFA), transforming growth factor beta (TGFB), insulin-like growth factor 2 (IGF2), and receptors for platelet-derived growth factor alpha (PDGFA), insulin, IGF1, and IGF2 during the pre-implantation stages of development. Basic fibroblast growth factor (FGFB) is a maternal transcript in cows, but the protein is detectable in bovine embryos until the 8-cell stage. IFNT mRNA is detectable in Day 8 bovine blastocysts, as are transcripts for insulin, epidermal growth factor (EGF), and nerve growth factor (NGF). Watson et al. [173] postulated that the pattern of gene expression is correlated with the timing of activation of the embryonic genome.

Supplementation of culture media for bovine embryos with various growth factors has been reported [174,175,176,177,178]. Thibodeaux et al. [177] suggested that embryotrophic effects of coculture systems are mediated via the secretion of growth factors. Platelet-derived growth factor (PDGF), present in oviductal secretions [174], induces bovine embryos to develop beyond the 8- to 16-cell stage in vitro by activating protooncogenes that decrease the duration of the fourth cell cycle [174,175]. BFGF and TGFA also accelerate progression through the fifth cell cycle [175], while TGFB acts synergistically with EGF to increase development of bovine embryos to the hatched blastocyst stage in vitro [176]. However, there are also reports of variable effects of growth factors and other factors on in vitro development of bovine embryos [178,179,180].

### 4.3. Elongation of Bovine Conceptuses and Progesterone

Bovine embryos develop to the blastocyst stage in vitro and in vivo, but only within the uterine environment do they elongate to the fully developed filamentous conceptus, because an undefined mechanism of communication between the blastocyst and uterine endometrium is required for growth and development of the conceptus and subsequent implantation and placentation during pregnancy. Embryonic development is accelerated in cows, with rapid increases in concentrations of P4 post-ovulation and final concentrations of P4 in blood [181,182,183,184,185]. The administration of exogenous P4 shortly after conception advances the timing of elongation of bovine and ovine conceptuses [182,184,185]. Progesterone induces the expression of many genes by the uterine epithelia of cows, some of which are expressed exclusively by the uterine LE and GE in response to P4 and IFNT [186,187]. In turn, genes and functions regulated by P4 and IFNT mediate specific changes in the uterine histotroph that govern conceptus survival and elongation [188,189]. The effect of P4 is mediated via the uterus to affect changes in unidentified embryotrophic factors within the uterine lumen, but there is no evidence of a direct effect on the conceptus to accelerate blastocyst development [167,183,188,190]. Larson et al. [191] failed to detect a direct effect of P4 from Day 1 to Day 3 or from Day 4 to Day 7 after fertilization on rates of embryonic development, and the addition of P4 to culture medium had no effect on conceptus elongation after transfer to synchronized recipients. In two other in vivo studies, increasing P4 did not influence blastocyst development. Carter et al. [185] reported that treatment of beef heifers with exogenous P4 from Day 3 had significant effects on elongation of conceptuses between Days 13 and 16 of pregnancy. But transfer of in vitro-produced embryos to the oviducts of beef heifers that did or did not receive a P4 insert on Day 3 after the onset of estrus had no effect on the proportion of embryos that developed to the blastocyst stage by Day 7 [192].

### 4.4. Implantation

Little is mechanistically known about the “Adhesion Cascade for Implantation” in cattle. The electron and light micrographic studies of King et al. [193,194] provide an excellent overview of the physical interactions between the blastocyst and uterus during this period. By Day 18 of gestation, the height of the uterine LE decreases as an initial response to the blastocyst. Attachment of the blastocyst trophectoderm to the uterine LE is first observed on Day 20, with attachment occurring simultaneously in caruncular and intercaruncular regions of the uterus near the embryo and then extending towards each end of the blastocysts. At this time, the microvilli on the uterine LE indent the apical membranes of the trophectoderm; this develops into interdigitation between the microvilli of both the trophectoderm and uterine LE by Day 24 and intimate attachment by Day 29 of gestation. Interestingly, large multinucleate cells are present in the uterine LE, and binucleate trophoblast giant cells (TGCs) are present in the early stages of implantation. Placentomes are first recognizable on Day 20 [193,194]. Alterations in the uterine LE have been reported to be limited to the formation of trinucleate syncytial cells formed through the fusion of TGCs with uterine LE cells [195], but recent immunofluorescent staining has shown that expanses of the uterine LE are temporally lost, and syncytialization is more extensive than previously realized [150,196].

## 5. Pregnancy in Mares

### 5.1. General (Figure 4)

Equine embryos remain in the oviduct for 6 to 7 days, where they develop to the late morula or early blastocyst stage with activation of the embryonic genome [197,198,199,200,201]. Only developing embryos are transported from the oviduct into the uterus, while unfertilized oocytes remain in the ampulla of the oviduct in mares [199,200]. Transport of developing embryos is attributed to secretion of PGE2 from Day 4 or 5 after fertilization, which induces the circular smooth muscle of the isthmus of the oviduct to relax and dilate, allowing the embryo to pass into the uterus [201,202,203].

Within the uterus, an acellular capsule of mucin-like glycoproteins forms between the trophectoderm and zona pellucida of the blastocyst. This may involve uterocalin secreted in response to P4, which binds fatty acids and retinol, perhaps to increase cross-linking of trophectoderm-derived glycoproteins [204,205,206,207,208]. A second capsule forms after the blastocyst hatches from the zona pellucida on Day 7 and increases in thickness until Day 11, when it has a bilaminar appearance due to additional uterine-derived glycoproteins [209,210]. Between Days 16 and 18, degeneration of the capsule begins and is completed between Days 20 and 30 of gestation in response to an unidentified factor(s) of either uterine or conceptus origin [209,210]. If luteolysis occurs before conceptus fixation on Day 16, degradation of the capsule does not occur [209,211]. Equine blastocysts that lack a capsule do not survive and establish pregnancies [212]. This may be because the capsule maintains integrity of the spherical conceptus and antiadhesive properties essential for migration throughout the uterine lumen until the capsule is lost and implantation/fixation occurs [212,213].

**Figure 4 animals-14-01882-f004:**
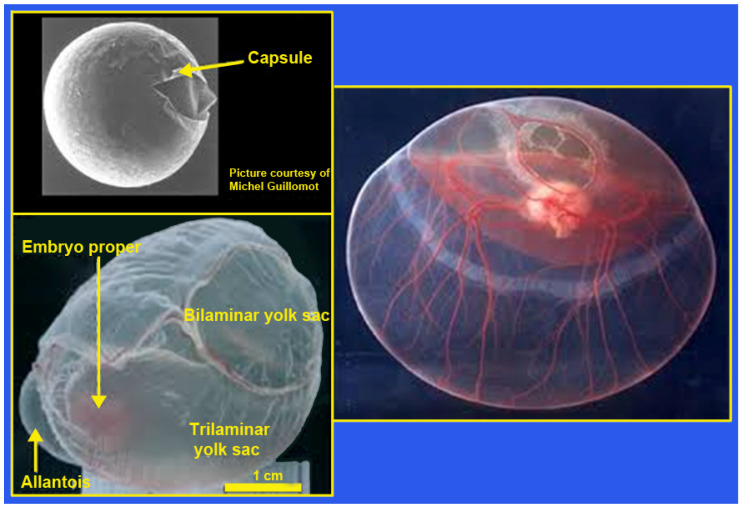
The equine conceptus; see [212]. This figure illustrates an equine blastocyst just before hatching from the capsule, membranes of the equine conceptus prior to implantation/fixation, and the equine conceptus with vascularization of the yolk sac and allantois.

### 5.2. Pregnancy Recognition Signaling

Embryonic signaling for maternal recognition of pregnancy is believed to occur as the conceptus migrates between the two uterine horns and contributes to transport and uptake of nutrients, binds endometrial proteins like uterocalin [214,215], and degrades B2M and soluble phospholipase A2 [210,216]. The conceptus is also a repository for proteins that likely affect development, including IGFBP3 and IGF1 [216]. The mechanism for pregnancy recognition in mares is not known, but the migrating conceptus is assumed to provide a biochemical signal to prevent luteolysis and ensure CL maintenance for production of P4 required for a uterine environment that supports its growth and survival [217]. Restricting migration of the conceptus to one uterine horn results in luteolysis and failure to establish pregnancy [218,219]. The presence of the conceptus inhibits pulsatile release of PGF in response to oxytocin from the posterior pituitary [220] or endometrium [221] between Days 10 and 16 after ovulation [222]. Pregnancy recognition signaling in mares is thought to begin on Day 12 of gestation [223] to inhibit expression of PTGS2 [224,225] and OXTR [226] during the period of conceptus migration, as the endometrium does not become responsive to oxytocin-induced secretion of PGF2 until after Day 16 of pregnancy [226,227], when there is also an increase in expression of PTGS2 and OXTR [228]. Equine conceptuses secrete IFND [229], PGE2 [230], IGF1 [231], and E2 [232] that may be involved in pregnancy recognition signaling, conceptus migration [219], angiogenesis and vasculogenesis [233], and uterine secretory activity [234]. There is differential expression of genes between pregnant and nonpregnant mares between Days 8 and 13 after ovulation and in response to E2 or PGE2 [235,236]; however, no gene product has been identified as a likely pregnancy recognition signal [223].

The equine conceptus produces an unknown factor(s) that inhibits uterine release of luteolytic PGF [228,237]. In cycling mares, concentrations of PGF in uterine venous plasma and uterine flushings increase between Days 14 and 16, when luteolysis occurs and concentrations of P4 in plasma decline. Receptors for PGF on luteal cells are abundant between Day 14 of the estrous cycle and estrus, as well as on Day 18 of pregnancy. The equine conceptus migrates between the two uterine horns until fixation on Day 18 of pregnancy, exerting an antiluteolytic mechanism as the amounts of PGF in uterine fluids and uterine venous plasma are reduced, and there is no pulsatile pattern of secretion of PGF in pregnant mares. The presence of the conceptus also abrogates endometrial production of PGF in response to both cervical stimulation via the Ferguson Reflex and exogenous OXT, indicating lack of expression of endometrial OXTR in pregnant mares. Equine conceptuses produce increasing amounts of E2 between Days 8 and 20 of gestation; however, attempts to prolong CL lifespan in mares by injections of E2 have yielded variable results [231]. However, administration of oxytocin prolongs luteal lifespan in mares [227], perhaps by downregulating expression of OXTR in uterine epithelia. The equine conceptus also secretes proteins of 400, 65, and 50 kDa between Days 12 and 14 of pregnancy, as well as IFND [229,231,238], but their role(s) in pregnancy recognition is not known.

### 5.3. Implantation

Loss of expression of PGR in uterine epithelia and loss of the blastocyst capsule is followed by attachment of the conceptus trophectoderm to the uterine LE, which induces proliferation of trophectoderm cells and modifies the uterine LE glycocalyx [239,240]. Mucin1 is not lost from the uterine LE in mares [241], as occurs in many mammals prior to implantation [239,240]. Rather, P4-dependent desialylation of the capsule allows the blastocyst to exhibit adhesive properties [210,213,223]. Progesterone from the CL and E2 from the conceptus likely stabilize attachment of the trophectoderm to the uterine LE, increasing expression of OPN and its receptors, CD44 and integrin αVβ3, as well as integrins and integrin-binding matrix proteins, such as fibronectin and fibrinogen [242].

Cytokines, growth factors, hormones, and their receptors are expressed to a greater extent by the trophectoderm, endometrium, or both between the third and fourth weeks of pregnancy in mares [223,242]. They include gene encoding for aromatase, PGE2, leukemia inhibitory factor and its receptor interleukin-6 signal transducer [243], members of the fibroblast growth factor family [223,243], and insulin-like growth factor binding proteins [244]. The uterine endometrium permits invasion of chorionic girdle cells and interdigitation of noninvasive trophectoderm/chorion and endometrial cells [245]. Factors adversely affecting implantation and/or maternal recognition of pregnancy signaling are associated with embryonic losses in mares between the time of conceptus fixation (Day 16) and the onset of placentation between Days 40 and 45 of gestation [242,246,247,248].

## 6. Conclusions

The events of early pregnancy are similar for sheep, pigs and cows but different from those for equine species; however, the outcome is to allow the conceptus intimate association with the maternal uterus for nutrients and gases required for development of the conceptus. The pregnancy recognition signaling mechanisms are clearly unique to pigs, as compared to sheep and cows and to horses, but the signaling in each case is to prevent regression of the CL required for production of P4 essential for establishment and maintenance of pregnancy. These events are followed by implantation, for which mechanisms differ somewhat among species, but the process stabilizes the relationship between the conceptus and uterine endometrium in preparation for development of a robust vascular exchange network required for greatly enhancing the efficient exchange of nutrients and gases between the fetal-placental and uterine vasculature.

## Figures and Tables

**Figure 1 animals-14-01882-f001:**
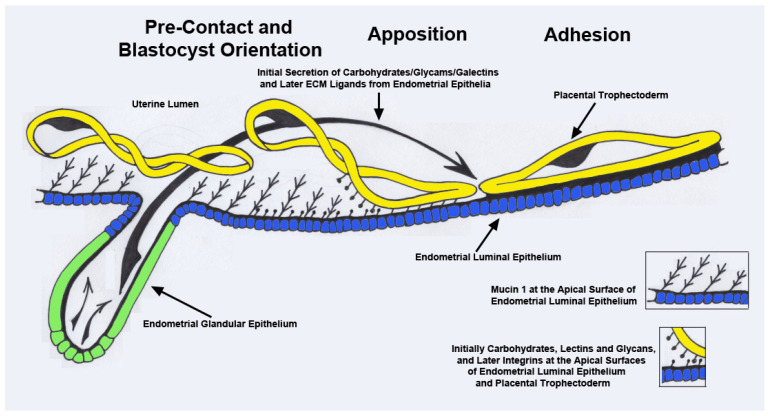
Illustration of the “Adhesion Cascade for Implantation” in pigs, sheep, and cattle immediately after hatching of the blastocyst from the zona pellucida and undergoing elongation (see Johnson [7,14]). The uterine luminal and glandular epithelia secrete and/or transport nutrients into the uterine lumen to support conceptus development.

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
