# Peer review of "Early Embryonic Development in Agriculturally Important Species"

_animals, 2024, doi:10.3390/ani14131882_

Round 1

Reviewer 1 Report

Comments and Suggestions for Authors

The MS submitted to Animals by Bazer and Johnson provides a very comprehensive review of the available literature describing early embryo development and implantation in pigs, sheep, cows and horses. The review is very complete and well written, easy to follow and with lot of interesting information regarding the embryo develoment in different specie. In my opinion the review will be a fundamental piece for researchers working in this field. Some minor comments are made below in order to improve the MS:

Abstract: Lines 20-25. This is a very long sentence, please re write it.

Introduction: line 48-49. Please check this recent paper by Zernicka-Goetz group (Junyent et al., 2024, Cell 187, 2838–2854). Maybe you can include some comment about this.

Line 109: delete "see 6" from the title. The same for all titles. It is better to include this info within the text.

Figure 2 is not very clear and the quality is poor. Please try to make a better figure and better legend to improve the MS. Same for Figure 3. Using BioRender or similar could be good.

Line 320-321: the sentence is not clear. Please rewrite it.

Author Response

The MS submitted to Animals by Bazer and Johnson provides a very comprehensive review of the available literature describing early embryo development and implantation in pigs, sheep, cows and horses. The review is very complete and well written, easy to follow and with lot of interesting information regarding the embryo develoment in different specie. In my opinion the review will be a fundamental piece for researchers working in this field. Some minor comments are made below in order to improve the MS:

Abstract: Lines 20-25. This is a very long sentence, please re write it.   
Response: Sentence modified.

Introduction: line 48-49. Please check this recent paper by Zernicka-Goetz group (Junyent et al., 2024, Cell 187, 2838–2854). Maybe you can include some comment about this.   
Response: Reference is not relevant.

Line 109: delete "see 6" from the title. The same for all titles. It is better to include this info within the text.   
Response: Change made.

Figure 2 is not very clear and the quality is poor. Please try to make a better figure and better legend to improve the MS. Same for Figure 3. Using BioRender or similar could be good.   
Response: The figure cannot be duplicated using Biorender.

Line 320-321: the sentence is not clear. Please rewrite it.  
Response: The sentence was modified.

Reviewer 2 Report

Comments and Suggestions for Authors

Dear authors, thank you for writing this important review about maternal recognition of pregnancy and implantation in these 4 livestock species.  The figures are very illustrative and help a lot to understand the processes from embryo development to implantation.

Below, please find some suggestions for minor changes to the article:

Line 28 remove "luminal"

Line 49/50 Instead of "ICM also called embryonic disc" refer to ICM as the  precursor of the embryonic disc. I suggest to use ICM for the pre-hatching blastocyst and embryonic disc for the elongating and filamentous form.

Line 81/82 and 88 You described above luminal epithelium as LE.  Please stay with LE for the rest of the article.

Line 310 (figure 2) Delete one of the two periods.

Line 468 Correct histotroph instead of histotoph.

Line 494 Write IFNAR1 and IFNAR2 instead of IFNAR1 and IFNAR1

Line 624 Changed "from" by form

Line 857 A discussion about the possible mechanism of the oxytocin action on the prolongation CL lifespan of mares would be appreciated

Author Response

Reviewer 2

Dear authors, thank you for writing this important review about maternal recognition of pregnancy and implantation in these 4 livestock species.  The figures are very illustrative and help a lot to understand the processes from embryo development to implantation.

Below, please find some suggestions for minor changes to the article:

Line 28 remove "luminal" 
Response: Change was made.

Line 49/50 Instead of "ICM also called embryonic disc" refer to ICM as the  precursor of the embryonic disc. I suggest to use ICM for the pre-hatching blastocyst and embryonic disc for the elongating and filamentous form.  
Response: THEY ARE THE SAME AS NOTED

Line 81/82 and 88 You described above luminal epithelium as LE.  Please stay with LE for the rest of the article. 
Response: OK, this was done.

Line 310 (figure 2) Delete one of the two periods. 
Response: OK, this was done.

Line 468 Correct histotroph instead of histotoph.
Response:  OK, this was done.

Line 494 Write IFNAR1 and IFNAR2 instead of IFNAR1 and IFNAR1 
Response: OK, correction made

Line 624 Changed "from" by form 
Response: OK, correction was made

Line 857 A discussion about the possible mechanism of the oxytocin action on the prolongation CL lifespan of mares would be appreciated  
Response: OK, an possible explanation was provided.

Reviewer 3 Report

Comments and Suggestions for Authors

The authors provide a detailed review of embryo elongation, conceptus attachment and pregnancy establishment in pigs, sheep, cattle, and horses. There is a great deal of information on the hormones, proteins, and genes found to be associated with these processes in each species, representing a useful source of knowledge. However, it is very text heavy, and its appeal would be improved by summarising large amounts of information in tables or providing more figures of signalling pathways. Most of my specific comments simply relate to typographical/minor errors, and I recommend adding some information from recent studies to a few sections.

L10: Delete “death”.

L21: Change “expression” to “express”.

L28: Delete “luminal”.

L59: Change “factors” to “factor”.

L91: Remove hyphens from “Never-the-less”.

L111-123: The requirement for two conceptuses in each uterine horn to establish pregnancy in the pig should be briefly mentioned here.

L159-170: The role of lipid metabolism and signalling at elongation as proposed by Walsh and colleagues (DOI: 10.1002/mrd.23553) should be briefly discussed here.

L320: Change “redirection” to “redirecting”.

L326: I suggest making the sentence more concise. Delete “and the lack of P4 required for establishment and maintenance of pregnancy” as this is already clear.

L494: Change duplicated “IFNAR1” to “IFNAR2”.

L511: Amend to “is localized”.

L522-523: Typos “ocytocin” and “prostagland”. Delete “in”.

L632: Change “decreased” to “decrease”.

L706: Please clarify what is meant by “CO-treated” ewes”. The CO abbreviation is not previously defined. Corn oil?

L838-860: Swegen and colleagues (2017) identified a number of proteins secreted by Day 10 equine embryos with potential roles in early pregnancy and speculated that production of prostaglandin F2α receptor inhibitor (PTGFRN) by the conceptus suppresses PGFR during early pregnancy in the mare. Also, the findings of Smits and colleagues (2018), who identified numerous proteins in the uterine fluid of pregnant mares, greatly coincide with the findings of Swegen et al. Relevant information from the studies of Swegen et al. (DOI: 10.1002/pmic.201600433) and Smits et al. (DOI:10.1038/s41598-018-23537-6) should be included here.

L887: The sentence doesn’t read quite right. Suggest replacing “its” with “maintain”.

Author Response

Reviewer 3

The authors provide a detailed review of embryo elongation, conceptus attachment and pregnancy establishment in pigs, sheep, cattle, and horses. There is a great deal of information on the hormones, proteins, and genes found to be associated with these processes in each species, representing a useful source of knowledge. However, it is very text heavy, and its appeal would be improved by summarising large amounts of information in tables or providing more figures of signalling pathways. Most of my specific comments simply relate to typographical/minor errors, and I recommend adding some information from recent studies to a few sections.

L10: Delete “death”. 
Response: OK, change was made

L21: Change “expression” to “express”. 
Response: OK, change was made

L28: Delete “luminal”. 
Response: OK, this was done

L59: Change “factors” to “factor”. 
Response: NO, factors is correct.

L91: Remove hyphens from “Never-the-less”. 
Response: OK, wording was changed.

L111-123: The requirement for two conceptuses in each uterine horn to establish pregnancy in the pig should be briefly mentioned here. 
Response: OK, this was added.

L159-170: The role of lipid metabolism and signalling at elongation as proposed by Walsh and colleagues (DOI: 10.1002/mrd.23553) should be briefly discussed here.

Response: The suggested reference is not relevant to this review.

L320: Change “redirection” to “redirecting”.  
Response: OK, this was done

L326: I suggest making the sentence more concise. Delete “and the lack of P4 required for establishment and maintenance of pregnancy” as this is already clear. 
Response: OK, the sentence was modified

L494: Change duplicated “IFNAR1” to “IFNAR2”. 
Response: OK, this was done

L511: Amend to “is localized”. 
Response: OK

L522-523: Typos “ocytocin” and “prostagland”. Delete “in”. 
Response: OK, corrections made.

L632: Change “decreased” to “decrease”.  
Response: OK, corrected.

L706: Please clarify what is meant by “CO-treated” ewes”. The CO abbreviation is not previously defined. Corn oil?  
Response: OK, this has been clarified in the review.

L838-860: Swegen and colleagues (2017) identified a number of proteins secreted by Day 10 equine embryos with potential roles in early pregnancy and speculated that production of prostaglandin F2α receptor inhibitor (PTGFRN) by the conceptus suppresses PGFR during early pregnancy in the mare. Also, the findings of Smits and colleagues (2018), who identified numerous proteins in the uterine fluid of pregnant mares, greatly coincide with the findings of Swegen et al. Relevant information from the studies of Swegen et al. (DOI: 10.1002/pmic.201600433) and Smits et al. (DOI:10.1038/s41598-018-23537-6) should be included here.

Response: These two references have been added.

L887: The sentence doesn’t read quite right. Suggest replacing “its” with “maintain”. 
Response:  OK, this has been done.

Reviewer 4 Report

Comments and Suggestions for Authors

See attached file for some minor comments on the manuscript. 

In general, I advise the Authors to consider introducing two minor structural improvements: 1) a list of abbreviations to facilitate fluency; 2) (if and where possible) further subdivide the manuscript into sub-paragraphs. 

It is a truly detailed and comprehensive review that contains a great amount of information even though it deals with a very limited topic, for the comparison between different species. I recommend publication after minor revision, no further review is needed from my side. 

Author Response

Reviewer 4

Lines 21 – expression to express 
Response: OK, change made.

Line 22 – Don’t understand comment 
Response: OK, this has been clarified.

Lines 49-50 – Revise sentenence 
Response: OK, this has been done.

Line 303 – spacing 
Response: OK, this has been done.

Line 310 – remove one period 
Response: OK, this has been done.

Reviewer 5 Report

Comments and Suggestions for Authors

This work provides a remarkable overview of our knowledge of the histo-physiological and biochemical mechanisms of early embryonic development. For me as a clinician, it has been relatively difficult to understand everything. Understanding implies a very substantial background of biochemical knowledge.

Allow me to make a few suggestions.

- A list of abbreviations and their meanings would be greatly appreciated.

- A summary table comparing the major differences or similarities between the species could enhance the interest of this work. 

- Would it not be possible to summarize the practical implications of this knowledge for each species?

105 Fig1 could you indicate more precisely the ICM on the figure ? Could you give some more informations on the role of the endometrial glandular and luminal epithelium ? Which part of the uterine epithelium is responsible of the synthesis of carbohydrates, lectines …

111 to 143 : any reference has been mentioned

124 to 170 : This paragraphy is rather difficult to understand without a figure

143 could you explain in some words the role of filopodia ?

254 what does IAC mean ?

280 and 414 and 546 What does OPN mean ?

305 Fig 2 What does KO aromatase and servomechanism mean ? How can we understand the left and and the right part of this figure ?

476 547 what does MTOR mean ? Line 420 you mention mechanistic target of rapamycin . Could you explain ?

817 what’s the difference between a bi and a trilaminar  yolk sac?  

825 B2M i.e ?

Author Response

Reviewer 5

This work provides a remarkable overview of our knowledge of the histo-physiological and biochemical mechanisms of early embryonic development. For me as a clinician, it has been relatively difficult to understand everything. Understanding implies a very substantial background of biochemical knowledge.

Allow me to make a few suggestions.

- A list of abbreviations and their meanings would be greatly appreciated.  
Response: Abbreviations have been explained thoughout the text when first used.

- A summary table comparing the major differences or similarities between the species could enhance the interest of this work.  
Response:  Such a table would be very complex and redundant since details of the differences and similarities have been discussed in the review.

- Would it not be possible to summarize the practical implications of this knowledge for each species?   
Response:  This review is to summarize basic knowledge of early pregancy that can be used in considering practival implications.  it is beyond the scope of this review to discuss practical implications of all that has been presented in this review.

105 Fig1 could you indicate more precisely the ICM on the figure ?  
Response: This can’t be done.

Could you give some more informations on the role of the endometrial glandular and luminal epithelium ? Which part of the uterine epithelium is responsible of the synthesis of carbohydrates, lectines …
Response: THIS HAS BEEN DONE

111 to 143 : any reference has been mentioned  
Response: The reference has been added.

124 to 170 : This paragraphy is rather difficult to understand without a figure  
Response: A figure is not available to address this point.

143 could you explain in some words the role of filopodia ?  
Response: Clarification added

254 what does IAC mean ? 
Response:  integrin adhesion complexes is stated followed by IAC.

280 and 414 and 546 What does OPN mean ?  
Response: Osteopontin as defined much earlier in the review.

305 Fig 2 What does KO aromatase and servomechanism mean ? How can we understand the left and and the right part of this figure ?  
Response: The left side of the figure shows the development of pig conceptuses on days 10, 11 and 15.  The right side of the figure describes key events required for conceptus development and establishment of pregnancy.   KO has been defined

476 547 what does MTOR mean ? Line 420 you mention mechanistic target of rapamycin . Could you explain ?   
Response: Mechanistic target of rapamycin is defined earlier in the review.

817 what’s the difference between a bi and a trilaminar  yolk sac?   
Response: Bilaminar is two apposing membranes and trilaminar is three apposing membranes.

825 B2M i.e ? 
Response: Beta 2 microglobulin as defined earlier in the review